# Anterior or Posterior Ankle Foot Orthoses for Ankle Spasticity: Which One Is Better?

**DOI:** 10.3390/brainsci12040454

**Published:** 2022-03-28

**Authors:** Carl P. C. Chen, Areerat Suputtitada, Watchara Chatkungwanson, Kittikorn Seehaboot

**Affiliations:** 1Department of Physical Medicine and Rehabilitation, Chang Gung Memorial Hospital at Linkou, College of Medicine, Chang Gung University, Guishan District, Taoyuan City 33343, Taiwan; carlchendr@gmail.com; 2Department of Rehabilitation Medicine, Faculty of Medicine, Chulalongkorn University, Bangkok 10330, Thailand; watchara_rehab@yahoo.com; 3Division of Rehabilitation Medicine, King Chulalongkorn Memorial Hospital, Bangkok 10330, Thailand; king.seecu@gmail.com; 4Excellent Center for Gait and Motion, King Chulalongkorn Memorial Hospital, Bangkok 10330, Thailand

**Keywords:** ankle foot orthosis (AFO), dynamic electromyography (dEMG), passive range of motion (PROM), Modified Ashworth Scale (MAS), spasticity

## Abstract

Background and Objectives: Ankle foot orthoses (AFOs) are commonly used by stroke patients to walk safely and efficiently. Both posterior AFOs (PAFOs) and anterior AFOs (AAFOs) are available. The objective of this study was to compare the efficacy of AAFOs and PAFOs in the treatment of ankle spasticity. Materials and Methods: A crossover design with randomization for the interventions and blinded assessors was used. Twenty patients with chronic stroke, a Modified Ashworth Scale (MAS) score of the ankle joint of 2, and a Tardieu angle ≥20 degrees were recruited. The patients were assigned to wear either an AAFO or PAFO at random and subsequently crossover to the other AFO. Results: Twenty stroke patients with ankle spasticity were recruited. The mean age was 46.60 (38–60) years. The mean time since stroke onset was 9.35 (6–15) months. It was discovered that the AAFO improved walking speed as well as the stretch reflex dynamic electromyography (dEMG) and walking dEMG amplitudes of the medial gastrocnemius muscles more significantly than the PAFO (*p* < 0.05). Conclusions: The AAFO had greater efficacy in reducing both static and dynamic ankle spasticity, and allowed for faster walking than the PAFO. The stretch reflex and walking dEMG amplitudes could be used for quantitative spasticity assessment.

## 1. Introduction

Hemiplegia is one of the most prevalent disabilities seen in the post-stroke phase and is characterized by asymmetrical gait abnormalities [1,2]. Muscle weakness can cause an asymmetrical gait, which can lead to ineffective movement, a loss of balance, and the risk of musculoskeletal injuries in healthy limbs [2]. Spasticity is a typical symptom of neurological impairment. It is commonly found after a stroke, causing difficulties with everyday activities and a lower quality of life [2]. Stroke patients frequently utilize ankle foot orthoses (AFOs) to assist in walking safely and efficiently. During the stance phase, both hinged and non-articulated AFOs provide mediolateral ankle stability as well as sufficient toe clearance during swing and heel-strike facilitation [1,2,3,4,5]. According to a recent meta-analysis, AFOs increased gait speed, cadence, step length, and stride length in stroke patients. Considering the sagittal plane angle formed by the ankle, knee, and hip was improved, patients with stroke who had ankle dorsiflexor weakness or hyper plantar flexion issues may benefit from an AFO. This meta-analysis provided basic information that can be utilized as a guide for administering AFOs to stroke patients in clinical practice [2]. AFOs come in a variety of styles, including adjustable metal and thermoplastic AFOs. For individuals with significant spastic inversion of the foot, metallic AFOs are preferred. However, most patients prefer thermoplastic orthoses because they are lighter and have a better cosmetic appearance [1,2,3,4,5]. The traditional posterior AFO (PAFO) is created using a lamination or vacuum-forming technique over a positive plaster model of the limb, commonly using thermoplastics with a posterior leaf-type design [1,2,3,4,5,6,7,8]. Anterior AFOs (AAFOs) are low-temperature or thermoplastic AFOs that are widespread in Asian countries because they are light, easy to use, and provide enhanced postural stability [9]. Prolonged stretching with a plastic ankle foot orthosis has also been used to alleviate spasticity [10]. AAFOs might help with gait characteristics, walking abilities, and balance control in stroke patients [9,11,12,13,14,15]. Studies that investigated the effects of AAFOs and PAFOs have been unable to conclude whether one style is better than the other [11,15]. AAFOs appear to be more effective in minimizing spasticity predominantly in the gastrocnemius and soleus muscles because there would be less stimulation of spasticity due to less direct contact of the plastic materials [9,11,12,13,14,15]. The ankle motion is less rigid when wearing an AAFO, leading to greater ankle motion and faster walking. This might be due to the anterior design featuring more ground reaction force that could assist the plantar flexor [9,11,12,13,14,15]. However, the superiority of AAFO remains inconclusive.

AFOs have been shown to improve mobility, walking speed, and balance. They have been used to help with mediolateral stability while standing, toe clearance during a swing, and heel strikes [1,2,3,4,5,6,10,15,16,17,18,19,20,21,22,23,24,25,26]. The positive effects of PAFOs on knee kinematics [20] and ankle kinematics [20,21,22] have also been described. As a result, the PAFO might have a favorable impact on the frequency of falls. The use of a PAFO has been shown to reduce fear of falling [13,23] and improve balance confidence [24]. When wearing PAFOs, higher AFO stiffness changes the ankle kinematics (lower peak ankle plantar flexion and increased dorsiflexion at the start), increases knee flexion at the initial contact, and reduces peak knee flexion and knee kinematics (i.e., increased knee flexion at initial contact, reduced peak knee flexion, and increased peak flexion and extension in the stance phase) [20,21,22,26]. The kinematic and clinical data for AAFOs remain scarce and inconclusive [9,11,12,13,14,15].

Electromyography (EMG) is used to predict post-stroke gait and rehabilitation therapy because it is sensitive to the neuromuscular abnormalities produced by stroke [27,28,29,30,31,32,33,34]. For the development of an EMG-based static and dynamic spasticity-monitoring system, we propose a portable EMG device or dynamic electromyography (dEMG), data processing, and data analytics for convenience in clinical practice. The dEMG is an objective spasticity measurement. The Modified Ashworth Scale shows a good relationship with the amplitude and duration of the dEMG response in the stretch reflex maneuver as a measure of static spasticity. The walking root mean square (RMS) amplitude and duration are said to be linked to dynamic spasticity when walking [27,28,29,30]. Some studies previously investigated the relationship between dEMG and muscle tension, and discovered that during voluntary contractions, dEMG and muscle tension exhibited a positive association [31,32,33]. Another study revealed that dEMG, combined with an isokinetic test, was a good quantitative method for measuring spasticity [30]. Since the MAS and Tardieu scale are more subjective and less quantitative, dEMG is considered the objective test of spasticity, which is highly accurate [27,28,29,30,31,32,33]. However, dEMG measurement necessitates the use of professionals and machines, and it is still underutilized in neuromuscular disease research. In the future, more modern dEMG machines that are simple to use and interpret should be available.

This is the first study to compare the effects of AAFOs and PAFOs on dEMG alterations in static and dynamic spasticity in stroke patients. The objective of this study was to determine whether AAFOs or PAFOs could alleviate ankle spasticity more effectively. Even though earlier data exist, this study will serve as a reference for future global research not only on the styles and types of AFOs but also on the use of dEMG for assessing gait, movement, and other neuromuscular activities.

The following are the contributions of this article:Description of the advantages of an anterior plastic AFO for use while barefoot at home over a posterior plastic AFO, which causes discomfort, imbalance, or increased spasticity, andIncorporation of dynamic EMG as a bedside or clinical tool for quantifying spasticity and other neuromuscular alterations, such as weakness.

## 2. Materials and Methods

### 2.1. Ethics and Clinical Trial Registry

The Institutional Review Board of the Faculty of Medicine, Chulalongkorn University, Bangkok, Thailand, approved the study (IRB no. 47951). Participants were informed about the procedure and provided written consent. The trial registration ID with the ICMJE-approved registry is TCTR20210225006.

### 2.2. Subjects

A randomized crossover clinical trial with a blinded assessor was performed to compare the effectiveness of anterior versus posterior plastic AFOs for the treatment of ankle spasticity in stroke patients. The inclusion criteria were as follows: diagnosis of unilateral hemiplegia caused by either hemorrhagic or ischemic stroke; ability to follow simple verbal commands or instructions; (3) Modified Ashworth Scale (MAS) score of the ankle joint of 2; (4) Tardieu angle of the ankle joint of ≥20 degrees; (5) no history of having worn an AFO before this study; and (6) ability to ambulate independently. Subjects were excluded if they had any of the following conditions: (1) medical problems other than stroke that would interfere with their gait; (2) foot-related premorbid or comorbid orthopedic problems; or (3) refusal to be enrolled or sign the informed consent form. All patients underwent neuroimaging studies, including computed tomography or magnetic resonance imaging of the brain, to confirm the diagnosis of early-stage stroke.

### 2.3. Procedure

Eligibility was determined for a total of 124 stroke patients. Figure 1 illustrates the Consolidated Standards of Reporting Trials (CONSORT). One hundred and four patients were excluded because they did not meet the inclusion criteria. The remaining 20 patients were randomly assigned to either anterior or posterior AFO treatment, as shown in Figure 2 and Figure 3. They applied each AFO for 30 days, 6 hours per day. The AFO was removed for a 1-week interval before crossover to the other. Every patient put on the same standard sandal shoes, secured with a Velcro strap. 

### 2.4. Plastic AFOs

Traditional PAFOs are made using a lamination or vacuum forming technique over positive plaster models of the limbs and have a posterior leaf-type design (Figure 3). Anterior AFOs, which have anterior leaf-type designs (Figure 2), are better for both barefoot and shoe-wearing walking indoors. Both anterior and posterior AFOs are made of thermoplastic materials, are rather rigid without hinge joints, and constantly cover the mediolateral ankle joint during walking.

### 2.5. Outcome Measurements

The outcome measurements were obtained before wearing each AFO and were evaluated at the end of 30 days of wear before being removed by a blinded assessor in terms of passive range of motion (PROM), the MAS score, walking velocity, and the stretch reflex dEMG and walking dEMG of the medial gastrocnemius muscle during a terminal stance phase. The root mean square (RMS) of the dEMG was used as the indicator and detected with wireless dEMG equipment (ME-6000, MEGA EMG^®^). Patient satisfaction was evaluated at the end of 30 days of wear before removing each AFO by a blinded assessor.

### 2.6. Wireless dEMG Equipment

The EMG was recorded with wireless dEMG equipment (ME-6000, MEGA EMG^®^), measured in microvolts (μV), with the capture and data processing software Megawin 3.0.1, manufactured by Mega Electronics Ltd. (Kuopio, Finland). The surface EMG was recorded using disposable, self-adhesive Ag/AgCl snap dual electrodes applied to the left and right sides of the participants’ leg muscles and connected to the wireless dEMG equipment via dEMG lead wires (Common Mode Rejection Ratio > 100 dB, Gain: 500). The electromyography data were filtered with a bandpass Butterworth filter at 6 Hz to remove noise from the dEMG signals, whose signal resolution was 14-bit after application of an analogue-to-digital converter (ADC). Artefacts such as physiological signals, low-frequency motion artefacts, power line sounds, and ADC clipping were manually checked on all recorded signals. The local grid’s 60 Hz AC noise was removed from the dEMG signal. Electrocardiography (ECG) signal artefacts were also removed from the dEMG signal [34]. To minimize impedance, the participants were given a low-alcohol swab to clean their skin. Considering the AFO prevents recording from numerous muscles, we only obtained dEMG data from the spastic leg’s medial gastrocnemius muscles, as shown in Figure 4. Before the testing, the participants were advised not to drink any caffeinated beverages or engage in any physical activity. Additionally, the subjects were allowed a short time to familiarize themselves with their prescribed AFOs. The gait speed was set at a comfortable level for each subject. To reduce errors in measurement during the dEMG recording, we calculated the mean of three repeated measurements. When gastrocnemius muscle spasticity was present, the dEMG activity was greater. Coherences in the frequency range of 20 to 100 Hz were used in this work to broadly cover the relevant range of the EMG signal. The most significant differences between healthy controls and patients with spasticity were found in this frequency range according to an examination of different frequency ranges. A previous study confirmed that the spasticity-induced stretch reflex of the antagonist muscle would lead to higher muscular coactivation between two antagonistic muscles such as, for example, the gastrocnemius and tibialis anterior [27,28,29,30,31].

### 2.7. dEMG Measurement

The mean and standard deviation of the dEMG measurements of the stretch reflex RMS amplitude of the gastrocnemius (μV) and the walking RMS amplitude of the gastrocnemius (μV) were calculated from three measurements [6,15,27,28,29,30,31]. An example of the dEMGs of the medial gastrocnemius are illustrated in Figure 5.

### 2.8. Statistical Analysis

All data were statistically analyzed using the Statistical Program for Social Sciences (SPSS) version 17 (SPSS Inc., Chicago, IL, USA). Descriptive statistics are displayed as the mean, range, and standard deviation. Normality was assessed using the Kolmogorov–Smirnov test. Intergroup comparisons were performed with the Mann–Whitney U test and comparisons within groups were assessed using the Wilcoxon signed-rank test. The level of significance used was *p* less than 0.05.

## 3. Results

Twenty stroke patients with ankle spasticity and a MAS score of two were recruited. Ten of the patients (50%) were male and ten (50%) were female. The mean age was 46.60 (38–60) years. The time since the onset of stroke was 9.35 (6–15) months. Twelve patients (60%) had ischemic strokes from an infarction and eight of them (40%) had hemorrhagic strokes. Ten (50%) were randomized to use an anterior AFOs first and the other ten (50%) were randomized to use the posterior AFOs first. Comparisons before and after using both types of AFOs revealed statistically significant improvements in passive range of motion (PROM), the Modified Ashworth Scale (MAS) score, walking velocity, stretch reflex, and walking dEMG of the medial gastrocnemius muscles during the terminal stance phase (*p* < 0.05), as shown in Table 1. When utilizing the AAFO, there was a statistically significant improvement over the use of the PAFO in the walking dEMG of the medial gastrocnemius muscles during the terminal stance phase (*p* < 0.05), as shown in Table 1. In addition, patient satisfaction was significantly higher when wearing an AAFO (*p* < 0.05), as shown in Table 2.

## 4. Discussion

In this study, we investigated the efficiency of two types of AFOs for reducing ankle spasticity, namely AAFOs and PAFOs, both of which are commonly used to support ankle weakness, and for stretching the ankle muscles. The clinical variables for measuring spasticity were the PROM of the ankle and the MAS score evaluated after 30 days of AFO use. According to the findings of the study, the two types of AFOs were shown to be equally effective in stretching the ankle muscles after 30 days of use.

The AAFO was observed to increase the patient’s walking speed more effectively than the PAFO. The patient’s front foot and heel may be able to contact the ground, which can also assist with ankle motion, while wearing the AAFO. There is no touch between the sole of the foot and the ground when wearing the PAFO; hence, there is no sensory feedback from the sole of the foot while walking. This result is in line with previous studies [11,15].

The surface EMG of the medial gastrocnemius muscle was used as the analytical measure in this study. The stretch reflex amplitude, which is closely related to the MAS score, and the root mean square (RMS) amplitude of the signal from this muscle while walking was measured [27,28,30,31,32,33]. The results demonstrated that the AAFO reduced both the static and dynamic spasticity of the ankle muscles after 30 days of wear, whereas the PAFO did not. The AAFO, as stated in our hypothesis, can reduce dynamic spasticity in this muscle during walking as well as static spasticity. The effects of AAFOs and PAFOs on changes in the dEMG in static and dynamic spasticity in stroke patients were compared for the first time in this study.

The AAFO has the potential to reduce static and dynamic ankle spasticity more than a PAFO. The AAFO might be more effective in cases with predominantly gastrocnemius spasticity since less spasticity was induced by contact with the plastic materials according to the dEMG changes in dynamic spasticity during walking. In addition, patient satisfaction was also higher while wearing the AAFOs. The dEMG of the stretch reflex and walking were recorded, and could be used for easy quantitative assessment of spasticity.

The dEMG measurement of the stretch reflex amplitude, which is related to the MAS score and RMS amplitude of muscle activity while walking, was used to assess dynamic spasticity [27,28,30,31,32,33]. Regarding the results of this study, dEMG demonstrated that the AAFO can reduce the dynamic spasticity of the gastrocnemius muscle while walking and increase walking speed more than PAFO. The PAFO could cause dynamic spasticity from contact with the agonist muscles while walking. There have been reports of spasticity being induced by stimulation of agonist muscles, the ball of the foot, or the palmar surface of the hand [2,9,15,27,28,29].

Furthermore, walking velocity was shown to be substantially higher while utilizing the AAFO. This could be the result of the ground reaction force following direct heel contact to the floor and the increased ankle control due to the dorsiflexion of the AAFO, allowing for heel striking and rapid midstance with toe lift. The anterior design is more similar to a reaction force AFO that assists the plantar flexor [9,11,12,13,15,18]. All of the individuals in this study had good heel strikes and push off, and faster walking with their AAFOs.

For static spasticity, the MAS score demonstrated a positive relationship with the amplitude and duration of the dEMG response in the stretch reflex maneuver [27,28,30,31,32,33]. The amplitude and duration of the walking root mean square (RMS) amplitude is said to be associated with dynamic spasticity when walking [27,28,30,31,32,33]. Our findings demonstrated a statistically significant decrement in the stretch reflex RMS amplitude and walking RMS amplitude of the gastrocnemius muscle when utilizing the AAFO, relative to not using the AFO, and when using the PAFO. In addition, there was an increase in the walking RMS amplitude when using the PAFO. Our findings validated the prediction that contact with plastic materials stimulated muscle spasticity, particularly while walking with greater shear stress when using the PAFOs [8,9,14,15].

Spasticity is defined as an increase in the RMS amplitude of the dEMG signal according to an increase in angular velocity and gait speed [28,30,31,32,33]. According to Xue Ping L et al. [30], pairing of the dEMG with an isokinetic test is a good quantitative method for assessing spasticity. In the clinic, dEMG and isokinetic evaluation could be utilized to measure muscular stiffness in stroke patients. For the instrumented assessment, the root mean square (RMS) amplitude of the electromyographic signal and torque were better predictors of a positive response by logistic regression analysis (area under the ROC curve = 0.82) when compared to the modified Tardieu angle (area under the ROC curve 0.7) [30]. Hu B. et al. [29] also employed the RMS amplitude of the stretch reflex dEMG signal for the objective evaluation of spasticity in detecting the stretch reflex onset (SRO). Our study was the first to compare the efficacy of the AAFO and PAFO by evaluating spasticity reduction using a novel method involving the stretch reflex and walking RMS amplitudes of the dEMG signal from the gastrocnemius muscles as an outcome measurement. Furthermore, our research showed that dEMG has greater sensitivity in measuring spasticity as it was able to identify a significant mean difference between the AAFO and PAFO before and after usage, whereas the PROM and MAS scores were not significantly different.

Increased computing power, wearable sensors, and mobile connectivity have ushered in plenty of mobile health innovations with the potential to transform stroke diagnosis and EMS response. By continually monitoring a broad range of biometric data, commercially available devices have the technological power to detect the cardinal verbal, motor, gait, and sensory manifestations of stroke. The next stage in stroke research is to intensify translational research in order to turn these technologies’ promises into validated, accurate, real-world deployments [35]. The result of this study gave the idea that dEMG can incorporate spasticity measurement.

According to a recent study, patients who received AFOs soon after stroke fell much more often than those who did not. However, it is worth noting that 63.6 percent of the falls in the first eight weeks of the study occurred in people who were not given an AFO. The remaining 36.4 percent of falls were in PAFO-wearers [25]. In the past, most AFO studies focused on the effects of PAFOs. Due to the hot weather, people in various Asian countries tend to walk around with bare feet indoors and the PAFO is not appropriate in such situations [9,11,12,13,15]. Chen CK et al. discovered that using an AAFO in the early stage of recovery may aid in postural stability [9]. The available evidence covering all AFO designs suggests that AFOs can minimize energy costs, increase weight transfer over the weak leg, and improve ankle and knee kinematics in hemiplegic stroke patients [18,19,20,21]. Many clinicians and patients, however, refuse to consider AFOs because of concerns about weight, pain, difficulty fitting into shoes, or appearance [19,21]. There is also concern about muscle disuse and delayed functional recovery during AFO use [19,21].

The following prospective new results show the potential disadvantages of the PAFO: (1) when worn with a shoe that does not have a rocker bottom, the solid AFO limited midfoot and forefoot movements by the foot plate, inhibiting the late stance phase, and (2) the plastic full heel cord and plantar surface covering triggered the muscle stretch reflex and increased gastrocnemius muscle activity as recorded by dEMG. On the other hand, patients who did not wear shoes with a proper rocker mechanism or had bare feet at home benefitted from AAFOs because their own feet could help with the late stance phase. The non-rigid ankle allowed for plantar flexion, which was assisted by the heel strike, resulting in faster walking. Our patients had never worn orthoses before, thus there was no possible prejudice (learning effect). In terms of walking comfort and spasticity reduction, AAFOs were statistically more satisfying than PAFOs.

### Limitations of the Study and Potential Implications

The multidimensional nature of spasticity indicates that there are different types of abnormal muscle activation during walking. The results of this study might be applicable only to the type and severity of spasticity in our included patients. Numerous studies have demonstrated the efficiency of PAFOs [1,2,3,4,5,6,7,8,9,11,15,17,18,19,20,21,23,24,25,26] and AAFOs [9,11,12,13,15]. Many elements, such as the diverse designs of the AFOs, the duration since stroke onset, the forms of spasticity (predominantly gastrocnemius or tibialis posterior spasticity), any co-contracture of the soft tissues, and so on, may need to be addressed in future studies. The dEMG could be used for quantitative spasticity assessment. The combination of kinematic analysis and balance analysis will provide more detail for the study of any design of AFO. However, it is more convenient and less expensive to conduct dEMG analysis alone. Furthermore, our research revealed that dEMG has greater sensitivity in measuring spasticity as it was able to identify a significant mean difference between the AAFO and PAFO before and after use, while the PROM and MAS scores were not significantly different.

The benefit of AFO for mild to moderate spasticity with a MAS score of two is that there are fewer side effects than with other treatments. AFOs assist in prolonging stretching. Botulinum toxin injections or alcohol neurolysis require professional skills, have a short duration of effect, and are costly. In addition, AAFOs could be used during prolonged stretching at home with bare feet comfortably, such as during a pandemic, and with less stimulation of spasticity in cases of predominantly gastrocnemius and soleus spasticity. However, AFOs cannot be used on their own in cases of severe spasticity that require botulinum toxin injection and/or neurolysis before AFO use.

This is, however, a prospective randomized crossover trial involving only one institution. Additional prospective RCT trials with a larger sample size and a longer follow-up time should be performed in an international multicenter study. More research is needed to establish the sensitivity of dEMG in measuring spasticity. Increased computing power, wearable sensors, and mobile connectivity have been launched in plenty of mobile health innovations that have the potential to improve stroke detection. Future studies should combine dEMG with cutting-edge technologies such as Automatic Acute Stroke Symptom Detection and Emergency Medical System Alerting via Mobile Health Technologies [35]. This study is unique and interesting since the AAFO is easy to produce and inexpensive. In addition, we introduce dEMG for accuracy of spasticity measurement that could be recommended in clinical practice due to portability, light weight, and ease of use by the clinician. This study should be used as a foundation for future global research into the accuracy of spasticity measurement.

## 5. Conclusions

In comparison to a PAFO, an AAFO can potentially minimize static and dynamic ankle spasticity, allowing for faster walking. The AAFO might be more effective for individuals with predominantly gastrocnemius spasticity since less spasticity is induced by contact of the plastic materials according to dEMG changes in dynamic spasticity during walking. In addition, patient satisfaction was also higher during the use of the AAFO. dEMG could be used to easily assess spasticity quantitatively and with higher sensitivity than the PROM and MAS scores.

## Figures and Tables

**Figure 1 brainsci-12-00454-f001:**
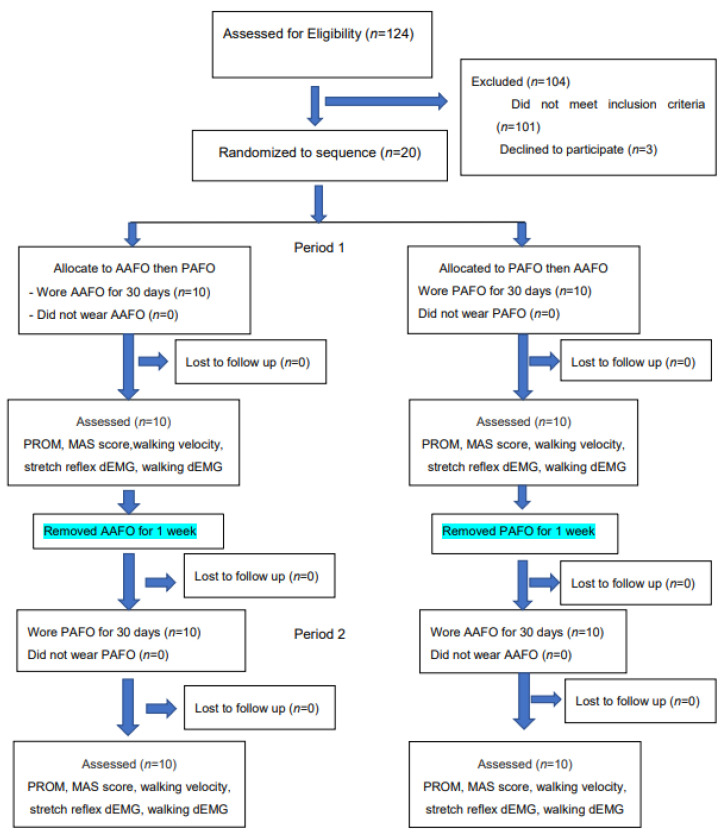
CONSORT flow chart for crossover study. Abbreviations: AAFO, anterior ankle foot orthosis; PAFO, posterior ankle foot orthosis; PROM, passive range of motion; MAS, Modified Ashworth Scale; dEMG, dynamic electromyography.

**Figure 2 brainsci-12-00454-f002:**
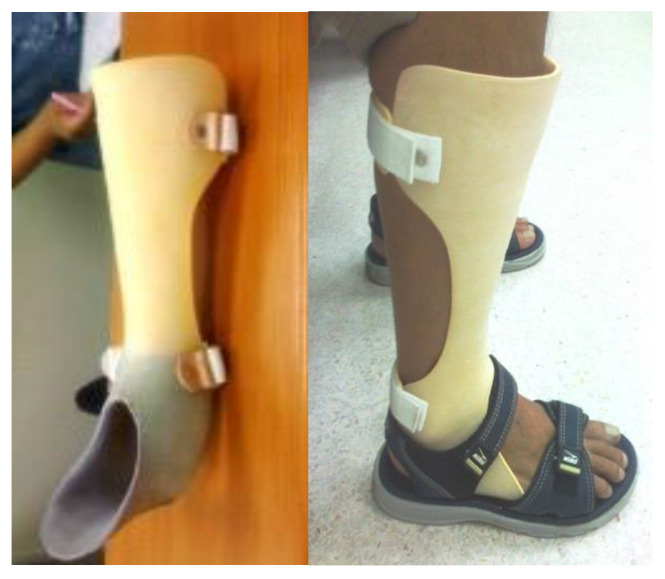
AAFO, anterior ankle foot orthosis.

**Figure 3 brainsci-12-00454-f003:**
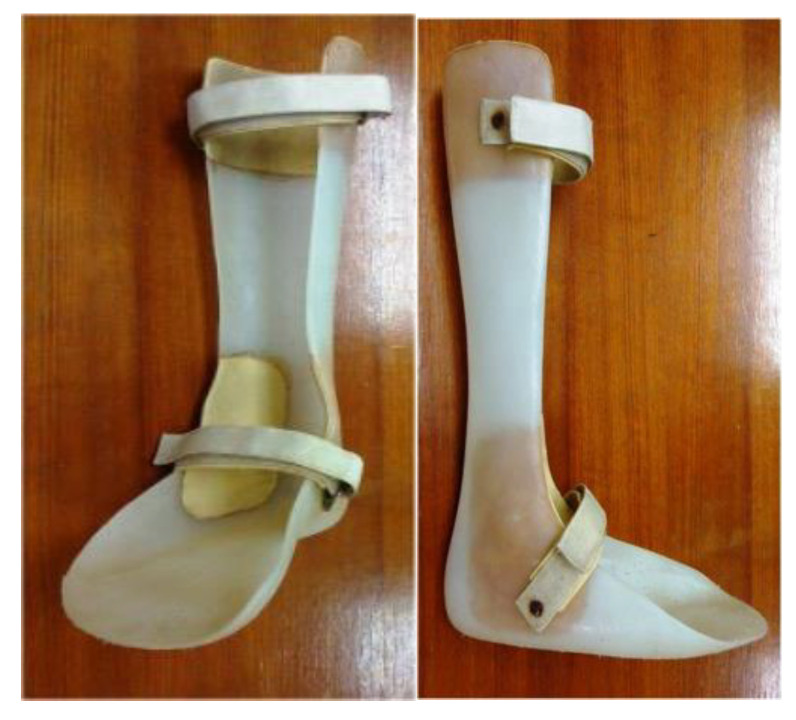
PAFO, posterior ankle foot orthosis.

**Figure 4 brainsci-12-00454-f004:**
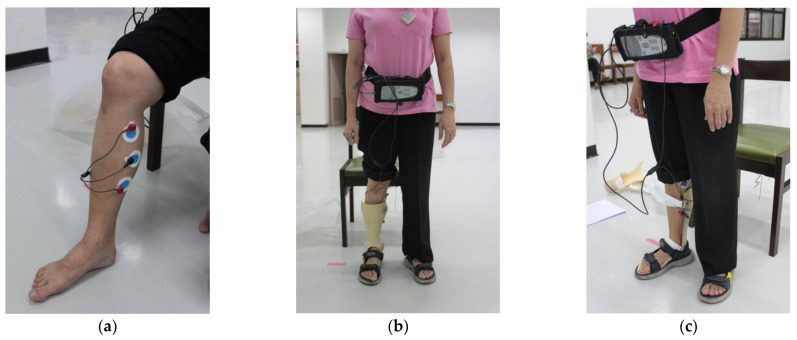
(**a**) dEMG from the medial gastrocnemius; (**b**) dEMG from the medial gastrocnemius while wearing an AAFO; and (**c**) dEMG from the medial gastrocnemius while wearing a PAFO. Abbreviations: AAFO, anterior ankle foot orthosis; PAFO, posterior ankle foot orthosis; dEMG, dynamic electromyography.

**Figure 5 brainsci-12-00454-f005:**
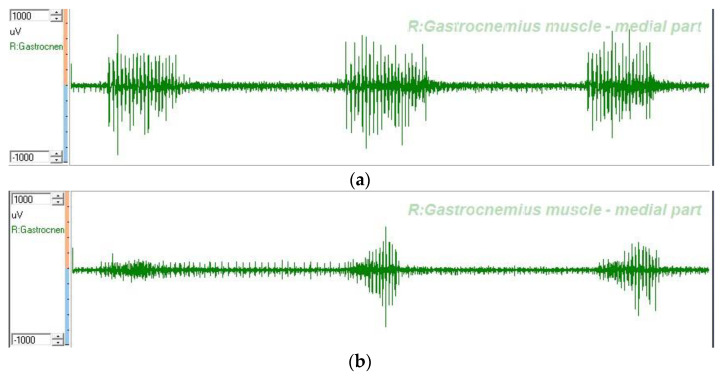
Depiction of dEMGs from right gastrocnemius-medial part. (**a**) Stretch reflex dEMG after wearing a PAFO. (**b**) Stretch reflex dEMG after wearing an AAFO. Abbreviations: R, right; AAFO, anterior ankle foot orthosis; PAFO, posterior ankle foot orthosis; dEMG, dynamic electromyography.

**Table 1 brainsci-12-00454-t001:** Comparison between AAFO and PAFO.

Parameter	Before AFO	After AFO	D	*p*	*p*
PROM of ankle DF (d)					
AAFO	3.00 ± 3.50	5.50 ± 3.70	2.50 ± 2.64	0.015 *	0.990
PAFO	3.00 ± 3.50	5.50 ± 3.69	2.50 ± 2.64	0.015 *	
MAS of GM					
AAFO	1.65 ± 0.24	1.30 ± 0.26	−0.35 ± 0.24	0.001 *	0.343
PAFO	1.65 ± 0.24	1.35 ± 0.34	−0.30 ± 0.26	0.005 *	
Walking velocity (m/s)					
AAFO	0.26 ± 0.21	0.28 ± 0.20	0.016 ± 0.017	0.015 *	0.021 **
PAFO	0.26 ± 0.21	0.27 ± 0.19	0.008 ± 0.025	0.343	
Stretch reflex RMS of GM (µV)					
AAFO	45.70 ± 33.59	37.60 ± 30.24	−8.10 ± 7.99	0.011 *	0.017 **
PAFO	51.20 ± 49.11	46.90 ± 46.59	−4.30 ± 5.33	0.311	
Walking RMS of GM (µV)					
AAFO	209.90 ± 233.87	146.80 ± 184.12	−62.50 ± 78.09	0.015 *	0.015 **
PAFO	197.60 ± 206.30	301.30 ± 298.72	103.70 ± 129.36	0.032 *	

Abbreviations: AFO, ankle foot orthosis; D, difference of mean data; PROM, passive range of motion; DF, dorsiflexion; d, degree; AAFO, anterior ankle foot orthosis; PAFO, posterior ankle foot orthosis; MAS, Modified Ashworth Scale; RMS, root mean square; GM, gastrocnemius muscle; * Mean difference comparisons within each group were performed with the Wilcoxon signed-rank test and the level of significance was *p* < 0.05. ** Mean difference comparisons between the AAFO and PAFO were performed with the Mann–Whitney U test and the level of significance was *p* < 0.05.

**Table 2 brainsci-12-00454-t002:** Comparison of patient satisfaction between the AAFO and PAFO.

Parameter	AAFO	PAFO	*p*
Beautiful	3.09 ± 0.30	2.55 ± 0.52	0.006 *
Strong and durable	2.91 ± 0.54	3.00 ± 0.00	0.341
Easy to put on	2.09 ± 0.55	2.73 ± 0.47	0.046 *
Comfort during walking	3.00 ± 0.00	2.54 ± 0.52	0.016 *
Reduces ankle spasticity	3.00 ± 0.00	2.27 ± 0.47	<0.001 *

Abbreviations: AAFO, anterior ankle foot orthosis; PAFO, posterior ankle foot orthosis; * Comparisons between the AAFO and PAFO were performed with the Mann–Whitney U test and the level of significance was *p* < 0.05.

## Data Availability

The data presented in this study are available upon request from the corresponding authors.

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
