# Peer review of "Anterior or Posterior Ankle Foot Orthoses for Ankle Spasticity: Which One Is Better?"

_brainsci, 2022, doi:10.3390/brainsci12040454_

Round 1

Reviewer 1 Report

In this cross-over design study, effects of AFO designs, anterior vs. posterior, on calf muscle spasticity were assessed in a group of stroke survivors. Results indicate that anterior AFO could reduce muscle spasticity during stretch reflex at rest and during walking than posterior AFO in case of gastrocnemius spasticity. Overall, the study included an adequate sample size with appropriate study design. Outcomes are clinically relevant and could be useful for guiding clinical practice. AAFOs could be used at home with bare feet comfortably during pandemic time and with less stimulation of spasticity in cases of predominantly gastrocnemius and soleus spasticity. The novel dynamic electromyography for objective measurement of spasticity which can used as the bedside or clinical setting is also interesting.

I think that it is well written and does a good job of providing hypothesis-generating research. 

However, I have 2 minor revision suggestions:

  1. In line 219 : The fact that 36.4 percent of PAFO patients still died [25]. I think it is wrong typo and should be: The fact that 36.4 percent of PAFO patients still fell down. [25]. 
  1. In Table 1, the normalization of RMS amplitude of Gastrocnemius (V) should be indicated for a clearer understanding of the dEMG value in spasticity.

Author Response

11 March 2022
Dear Editor and Reviewer 1,

We thank you and the reviewers for the comments and suggestions. Please find our point-by-point responses below.
We hope that our responses are satisfactory. Should there be anything that might improve our work, please kindly inform us. Thank you very much for your kind consideration.

Best Regards,

Professor Dr. Areerat Suputtitada, MD.
On behalf of the authors

Response to Reviewer 1 Comments
In this cross-over design study, effects of AFO designs, anterior vs. posterior, on calf muscle spasticity were assessed in a group of stroke survivors. Results indicate that anterior AFO could reduce muscle spasticity during stretch reflex at rest and during walking than posterior AFO in case of gastrocnemius spasticity. Overall, the study included an adequate sample size with appropriate study design. Outcomes are clinically relevant and could be useful for guiding clinical practice. AAFOs could be used at home with bare feet comfortably during pandemic time and with less stimulation of spasticity in cases of predominantly gastrocnemius and soleus spasticity. The novel dynamic electromyography for objective measurement of spasticity which can used as the bedside or clinical setting is also interesting.
I think that it is well written and does a good job of providing hypothesis-generating research. 
However, I have 2 minor revision suggestions:
Authors : We thank you and the reviewers for the comments and suggestions. Please find our point-by-point responses below.
We hope that our responses are satisfactory. Should there be anything that might improve our work, please kindly inform us. Thank you very much for your kind consideration.

1.    In line 219 : The fact that 36.4 percent of PAFO patients still died [25]. I think it is wrong typo and should be: The fact that 36.4 percent of PAFO patients still fell down. [25]. 
Response 1: Line 349-350 

The remaining 36.4 percent of falls were in PAFO wearers. [25]
2.    In Table 1, the normalization of RMS amplitude of Gastrocnemius (V) should be indicated for a clearer understanding of the dEMG value in spasticity.
Response 2: Table 1 
         Normalization RMS amplitude of the gastrocnemius (µV) 57.00 ± 39.00

Reviewer 2 Report

This study aimed to assess the efficacy of AAFOs and PAFOs for the treatment of ankle spasticity.  I have the following major suggestions.

  1. Please add a paragraph about the contribution of this article in a bulleted form at the end part of the Introduction section.
  2. Which novelty do authors claim for this article?
  3. Why did authors use only medial gastrocnemius muscle for their study?
  4. Authors should review gait/muscular changes due to diseases, fatigue, physical trauma and improve references mentioning studies of various neuromuscular changes in the article, Prediction of Myoelectric Biomarkers in Post-Stroke Gait.
  5. EMG is highly sensitive to the powerline and cardiac artifacts. In EMG data preprocessing, authors need to mention how you handle AC power and ECG artifacts in EMG signals.
  6. Introduction section needs to be more extended and improved. Authors should improve references mentioning studies related to gait changes due to diseases, such as Real-time Gait Monitoring System for Consumer Stroke Prediction Service
  7. Authors should add a figure of the experimental protocol used in this study.
  8. Authors should add more details in the captions of the figures.
  9. Authors should provide an error-bar plot of Table 1 for better visualization.
  10. Authors should discuss the strength and weaknesses of reported findings with other previous findings in the discussion section.
  11. From the writing point of view, the manuscript must be checked for typos and the grammatical issues should be improved.

Author Response

11 March 2022
Dear Editor,

We thank you and the reviewers for the comments and suggestions. Please find our point-by-point responses below.
We hope that our responses are satisfactory. Should there be anything that might improve our work, please kindly inform us. Thank you very much for your kind consideration.

Best Regards,

Professor Dr. Areerat Suputtitada, MD.
On behalf of the authors

Response to Reviewer 2 Comments
This study aimed to assess the efficacy of AAFOs and PAFOs for the treatment of ankle spasticity.  I have the following major suggestions.
Authors: We thank you and the reviewers for the comments and suggestions. Please find our point-by-point responses below. We hope that our responses are satisfactory. Should there be anything that might improve our work, please kindly inform us. Thank you very much for your kind consideration.
1.    Please add a paragraph about the contribution of this article in a bulleted form at the end part of the Introduction section.

Response 1: Line 97-101 
The following are the contributions of this article:
1. Description of the advantages of an anterior plastic AFO for use while barefoot at home over a PAFO, which causes discomfort, imbalance, or increased spasticity.
2. Incorporation of dynamic EMG as a bedside or clinical tool for quantifying spasticity and other neuromuscular alterations, such as weakness.

2.    Which novelty do authors claim for this article?
Response 2:  Line 94-96
Even though earlier data exist, this study will serve as a reference for future global research, not only on the styles and types of AFOs but also on the use of dEMG for assessing gait, movement, and other neuromuscular activities.

3.    Why did authors use only medial gastrocnemius muscle for their study?
Response 3: Line 215-217
Because the AFO prevents recording from numerous muscles, we only obtained EMG data from the spastic leg's medial gastrocnemius muscles, as shown in Figure 4.

4.    Authors should review gait/muscular changes due to diseases, fatigue, physical trauma and improve references mentioning studies of various neuromuscular changes in the article, Prediction of Myoelectric Biomarkers in Post-Stroke Gait.
Response 4: Line 47-51
Hemiplegia is one of the most prevalent disabilities seen in the poststroke phase and is characterized by asymmetrical gait abnormalities. Muscle weakness can cause an asymmetrical gait, which can lead to ineffective movement, a loss of balance, and the risk of musculoskeletal injuries in healthy limbs. Spasticity is a typical symptom of neurological impairment. It is commonly found after a stroke, causing difficulties with everyday activities and a lower quality of life.

Line 82-91
Electromyography (EMG) is used to predict poststroke gait and rehabilitation therapy because it is sensitive to neuromuscular abnormalities produced by stroke. [27-34] For the development of an EMG-based static and dynamic spasticity monitoring system, we proposed a portable EMG device or dynamic electromyography (dEMG), data processing, and data analytics for convenience in clinical practice. The dEMG is an objective spasticity measurement. The Modified Ashworth Scale shows a good relationship with the amplitude and duration of the dEMG response in the stretch reflex manoeuvre as a measure of static spasticity. The walking root mean square (RMS) amplitude and duration are said to be linked to dynamic spasticity when walking. [27-30] Some studies previously investigated the relationship between dEMG and muscle tension and discovered that during voluntary contractions, dEMG and muscle tension exhibited a positive association. [31-33] Another study revealed that dEMG, combined with an isokinetic test, was a good quantitative method for measuring spasticity.[30]
34.  Hussain I, Park SJ. Prediction of Myoelectric Biomarkers in Post-Stroke Gait. Sensors (Basel). 2021 Aug 7;21(16):5334. doi: 10.3390/s21165334.

5.    EMG is highly sensitive to the powerline and cardiac artifacts. In EMG data preprocessing, authors need to mention how you handle AC power and ECG artifacts in EMG signals.
Response 5: Line 203-218
Wireless dEMG equipment
The EMG was recorded with wireless dEMG equipment , MEGA ME6000 MT-M6T8-0-10 [22], measured in microvolts (μV), with the capture and data processing software Megawin 3.0.1, manufactured by Mega Electronics Ltd. (Kuopio, Finland). The surface EMG was recorded using disposable, self-adhesive Ag/AgCl snap dual electrodes applied to the left and right sides of the participants' leg muscles and connected to the wireless ME-6000, MEGA EMG® sensor via EMG lead wires (Common Mode Rejection Ratio > 100 dB, Gain: 500). The electromyography data were filtered with a bandpass Butterworth filter at 6 Hz to remove noise from the dEMG signals, whose signal resolution was 14-bit after application of an analogue-to-digital converter (ADC). Artefacts such as physiological signals, low-frequency motion artefacts, power line sounds, and ADC clipping were manually checked on all recorded signals. The local grid's 60 Hz AC noise was removed from the EMG signal. Electrocardiography (ECG) signal artefacts were also removed from the EMG signal [34]. To minimize impedance, the participants were given a low-alcohol swab to clean their skin. Because the AFO prevents recording from numerous muscles, we only obtained EMG data from the spastic leg's medial gastrocnemius muscles, as shown in Figure 4. Before the testing, the participants were advised not to drink any caffeinated beverages or engage in any physical activity.

6.    Introduction section needs to be more extended and improved. Authors should improve references mentioning studies related to gait changes due to diseases, such as Real-time Gait Monitoring System for Consumer Stroke Prediction Service
Response 6: Line 337-346
We add in the discussion 
Furthermore, our research showed that dEMG has greater sensitivity in measuring spasticity, as it was able to identify a significant mean difference between the AAFO and PAFO before and after usage, whereas the PROM and MAS scores were not significantly different.
Increased computing power, wearable sensors, and mobile connectivity have ushered in plenty of mobile health innovations with the potential to transform stroke diagnosis and EMS response. By continually monitoring a broad range of biometric data, commercially available devices have the technological power to detect the cardinal verbal, motor, gait, and sensory manifestations of stroke. The next stage in stroke research is to intensify translational research in order to turn these technologies' promises into validated, accurate real-world deployments. [35] The result of this study gave the idea that dEMG can incorporate spasticity measurement.
35. Bat-Erdene BO, Saver JL. Automatic Acute Stroke Symptom Detection and Emergency Medical Systems Alerting by Mobile Health Technologies: A Review. J Stroke Cerebrovasc Dis. 2021 Jul;30(7):105826. doi: 10.1016/j.jstrokecerebrovasdis.2021.105826.

7.    Authors should add a figure of the experimental protocol used in this study.
Response 7: Line 146-184
The Figure 1 was adjusted for more clarification of the experimental protocol. 
Figure 1 CONSORT flow chart for crossover study

                                      Period 1

                                      Period 2

8.    Authors should add more details in the captions of the figures.
Response 8 :  Figure 4 were added for more details of Figure 2,3 

The captions of the figures were added more details.

9.    Authors should provide an error-bar plot of Table 1 for better visualization.
Response 9 : The table1 was adjusted to be better visualized 

10.    Authors should discuss the strength and weaknesses of reported findings with other previous findings in the discussion section.

       Response 10: Line 285-302
4. DISCUSSION
11.    In this study, we investigated the efficiency of two types of AFOs for reducing ankle spasticity, AAFOs and PAFOs, both of which are commonly used to support ankle weakness, and for stretching the ankle muscles. The clinical variables for measuring spasticity were the PROM of the ankle and the MAS score evaluated after 30 days of AFO use. According to the findings of the study, the two types of AFOs were shown to be equally effective in stretching the ankle muscles after 30 days of use.
12.    The AAFO was observed to increase the patient's walking speed more effectively than the PAFO. The patient's front foot and heel may be able to contact the ground, which can also assist with ankle motion, while wearing the AAFO. There is no touch between the sole of the foot and the ground when wearing the PAFO; hence, there is no sensory feedback from the sole of the foot while walking. This result is in line with previous studies. [11,15]
13.    The surface EMG of the medial gastrocnemius muscle was used as the analytical measure in this study. The stretch reflex amplitude, which is closely related to the MAS score, and the root mean square (RMS) amplitude of the signal from this muscle while walking were measured. [27,28,30-33] The results demonstrated that the AAFO reduced both static and dynamic spasticity of the ankle muscles after 30 days of wear, whereas thee PAFO did not. The AAFO, as stated in our hypothesis, can reduce dynamic spasticity in this muscle during walking as well as static spasticity. The effects of AAFOs and PAFOs on changes in the dEMG in static and dynamic spasticity in stroke patients were compared for the first time in this study.

11.    From the writing point of view, the manuscript must be checked for typos and the grammatical issues should be improved.

Response 11: The manuscript was edited and submitted for the editing service in English language from Springer Nature with a certificate attached.

Reviewer 3 Report

The study compare between two types of AFO on small sample of chronic and mild stroke. The research idea is good. However, the reporting was insufficent, and there were significant shortcoming in the study methodology. 

Abstract: 
- Line 15: '' There was evidence of posterior AFO (PAFO) and anterior AFO (AAFO)". The sentence is not clear 

- The objective of the study should be to compare between the effect of AAFO and PAFO

Introduction: 
- The introduction is very hard to follow
- There are many English mistakes 
- Please cite the references correctly in the same sentence not in a new sentence after the dot. 
- Please Justify the rationale of your study 
- Please mention a summary about the current evidence and what is your study add? 

Methods: 
- Please follow CONSORT statement of reporting cross-over RCTs
- How the study consider the carry-over effect? 

Discussion: 
- What about the generalization of your findings? The inclusion criteria were for mild stroke (Walking independently), and chronic stroke (More than six months) 
- What are the research recommendations. 

Author Response

11 March 2022
Dear Editor and reviewer 3

We thank you and the reviewers for the comments and suggestions. Please find our point-by-point responses below.
We hope that our responses are satisfactory. Should there be anything that might improve our work, please kindly inform us. Thank you very much for your kind consideration.

Best Regards,

Professor Dr. Areerat Suputtitada, MD.
On behalf of the authors

Response to Reviewer 3 Comments
The study compare between two types of AFO on small sample of chronic and mild stroke. The research idea is good. However, the reporting was insufficent, and there were significant shortcoming in the study methodology. 
 Authors: We thank you and the reviewers for the comments and suggestions. Please find our point-by-point responses below. We hope that our responses are satisfactory. Should there be anything that might improve our work, please kindly inform us. Thank you very much for your kind consideration.
Response 1: The manuscript was edited and submitted for the editing service in English language from Springer Nature with a certificate attached.
Abstract: 
- Line 15: '' There was evidence of posterior AFO (PAFO) and anterior AFO (AAFO)". The sentence is not clear
Response 2: Line 51-60 

Stroke patients frequently utilize ankle foot orthoses (AFOs) to assist in walking safely and efficiently. During stance phase, both hinged and nonarticulated AFOs provide mediolateral ankle stability, as well as sufficient toe clearance during swing and heel strike facilitation. [1-5] According to a recent meta-analysis, AFOs increased gait speed, cadence, step length, and stride length in stroke patients. Because the sagittal plane angle formed by the ankle, knee, and hip was improved, patients with stroke who had ankle dorsiflexor weakness or hyper plantar flexion issues may benefit from an AFO. This meta-analysis provided basic information that can be utilized as a guide for administering AFOs to stroke patients in clinical practice. [2] AFOs came in a variety of styles, including adjustable metal and thermoplastic AFOs. For individuals with significant spastic inversion of the foot, metallic AFOs are preferred. However, most patients prefer thermoplastic orthoses because they are lighter and have a better cosmetic appearance. [1-5]
- The objective of the study should be to compare between the effect of AAFO and PAFO
 Response 3: Line 27-28
The objective of this study was to compare the efficacy of AAFOs and PAFOs in the treatment of ankle spasticity.
Introduction: 
- The introduction is very hard to follow
- There are many English mistakes    I have rewrite and submitted for the editing service in English language from Springer Nature with a certificate attached. 
- Please cite the references correctly in the same sentence not in a new sentence after the dot. 
- Please Justify the rationale of your study 
- Please mention a summary about the current evidence and what is your study add? 
Response 4: Line 92-101 
This is the first study to compare the effects of AAFOs and PAFOs on dEMG alterations in static and dynamic spasticity in stroke patients. The objective of this study was to determine whether AAFOs or PAFOs could alleviate ankle spasticity more effectively. Even though earlier data exist, this study will serve as a reference for future global research, not only on the styles and types of AFOs but also on the use of dEMG for assessing gait, movement, and other neuromuscular activities.
The following are the contributions of this article:
1. Description of the advantages of an anterior plastic AFO for use while barefoot at home over a PAFO, which causes discomfort, imbalance, or increased spasticity.
2. Incorporation of dynamic EMG as a bedside or clinical tool for quantifying spasticity and other neuromuscular alterations, such as weakness.

Methods: 
- Please follow CONSORT statement of reporting cross-over RCTs
Response 5: Line 146-184
Figure 1 CONSORT flow chart for crossover study

                                      Period 1

                                      Period 2

- How the study consider the carry-over effect? 
Response 6: Line 122-123 , Line 169-170 in Figure 1

The AFO was removed for a 1-week interval before crossover to the other device.

Discussion: 
- What about the generalization of your findings? The inclusion criteria were for mild stroke (Walking independently), and chronic stroke (More than six months) 
Response 7: Line 367-391 
Limitations of the study and potential implications
The multidimensional nature of spasticity indicates that there are different types of abnormal muscle activation during walking. The results of this study might be applicable only to the type and severity of spasticity in our included patients. Numerous studies have demonstrated the efficiency of PAFOs. [1-9, 11, 15, 17-21, 23-26] and AAFOs. [9, 11-13, 15] Many elements, such as the diverse designs of the AFOs, the duration since stroke onset, the forms of spasticity (predominantly gastrocnemius or tibialis posterior spasticity), any co-contracture of the soft tissues, and so on, may need to be addressed in future studies. The dEMG could be used for quantitative spasticity assessment. The combination of kinematic analysis and balance analysis will provide more detail for the study of any design of AFO. However, it is more convenient and less expensive to conduct dEMG analysis alone. Furthermore, our research revealed that dEMG has greater sensitivity in measuring spasticity, as it was able to identify a significant mean difference between the AAFO and PAFO before and after use, while the PROM and MAS score were not significantly different.
The benefit of AFO for mild to moderate spasticity with a MAS score of 2 is that there are fewer side effects than with other treatments. AFOs assist in prolonging stretching. Botulinum toxin injections or alcohol neurolysis require professional skills, have a short duration of effect and are costly. In addition, AAFOs could be used during prolonged home stay with bare feet comfortably, such as during a pandemic, and with less stimulation of spasticity in cases of predominantly gastrocnemius and soleus spasticity. However, AFOs cannot be used on their own in cases of severe spasticity that require botulinum toxin injection and/or neurolysis before AFO use.
This is, however, a prospective randomized crossover trial involving only one institution. Additional prospective RCT trials with a larger sample size and a longer follow-up time should be performed in an international, multicentre environment. More research is needed to establish the sensitivity of dynamic electromyography in measuring spasticity. This study is unique and interesting since the AAFO is easy to produce and inexpensive, and dynamic electromyography devices inexpensive machines that should be recommended in clinical practice due to their portability, light weight, and ease of use by the clinician. This study could be used as a foundation for future global research.

- What are the research recommendations. 
Response 8: Line 379-391 

The benefit of AFO for mild to moderate spasticity with a MAS score of 2 is that there are fewer side effects than with other treatments. AFOs assist in prolonging stretching. Botulinum toxin injections or alcohol neurolysis require professional skills, have a short duration of effect and are costly. In addition, AAFOs could be used during prolonged home stay with bare feet comfortably, such as during a pandemic, and with less stimulation of spasticity in cases of predominantly gastrocnemius and soleus spasticity. However, AFOs cannot be used on their own in cases of severe spasticity that require botulinum toxin injection and/or neurolysis before AFO use.
This is, however, a prospective randomized crossover trial involving only one institution. Additional prospective RCT trials with a larger sample size and a longer follow-up time should be performed in an international, multicentre environment. More research is needed to establish the sensitivity of dynamic electromyography in measuring spasticity. This study is unique and interesting since the AAFO is easy to produce and inexpensive, and dynamic electromyography devices inexpensive machines that should be recommended in clinical practice due to their portability, light weight, and ease of use by the clinician. This study could be used as a foundation for future global research.
Line 393-399 
5. CONCLUSIONS
In comparison to a PAFO, an AAFO can potentially minimize static and dynamic ankle spasticity, allowing for faster walking. The AAFO might be more effective for individuals with predominantly gastrocnemius spasticity since less spasticity is induced by contact of the plastic materials according to dEMG changes in dynamic spasticity during walking. In addition, patient satisfaction was also higher during the use of the AAFO. dEMG could be used to easily assess spasticity quantitatively and with higher sensitivity than the PROM and MAS score.

Round 2

Reviewer 2 Report

Thanks for addresing the comments. 

Author Response

14 March 2022
Dear Editor and Reviewer 2,
We thank you and the reviewers for the comments and suggestions. Please find our editing certificate in the attached file  
The manuscript was edited and submitted for the editing service in English language from Springer Nature with a certificate attached.
We hope that our responses are satisfactory. Should there be anything that might improve our work, please kindly inform us. Thank you very much for your kind consideration.

Best Regards,

Professor Dr. Areerat Suputtitada, MD.
On behalf of the authors

Reviewer 3 Report

Thank you for revising the manuscript. It was improved. Please find my follow comments: 

  • The following sentences need supportive references

    ''Hemiplegia is one of the most prevalent disabilities seen in the poststroke phase and is characterized by 47 asymmetrical gait abnormalities. Muscle weakness can cause an asymmetrical gait, which can lead to ineffective 48 movement, a loss of balance, and the risk of musculoskeletal injuries in healthy limbs. Spasticity is a typical 49 symptom of neurological impairment. It is commonly found after a stroke, causing difficulties with everyday 50 activities and a lower quality of life ''.
  • Please link the followinf paragraph with the limitation in the current evidence: 
    '' Electromyography (EMG) is used to predict poststroke gait and rehabilitation therapy because it is sensitive to 82 neuromuscular abnormalities produced by stroke. [27-34] For the development of an EMG-based static and 83 dynamic spasticity monitoring system, we proposed a portable EMG device or dynamic electromyography 84 (dEMG), data processing, and data analytics for convenience in clinical practice. The dEMG is an objective 85 spasticity measurement. The Modified Ashworth Scale shows a good relationship with the amplitude and duration 86 of the dEMG response in the stretch reflex manoeuvre as a measure of static spasticity. The walking root mean 87 square (RMS) amplitude and duration are said to be linked to dynamic spasticity when walking. [27-30] Some 88 studies previously investigated the relationship between dEMG and muscle tension and discovered that during 89 voluntary contractions, dEMG and muscle tension exhibited a positive association. [31-33] Another study revealed 90 that dEMG, combined with an isokinetic test, was a good quantitative method for measuring spasticity.[30]''

- The manuscript is still very long and hard to follow. 

Author Response

14 March 2022
Dear Editor and Reviewer 3,

We thank you and the reviewers for the comments and suggestions. Please find our point-by-point responses below.
We hope that our responses are satisfactory. Should there be anything that might improve our work, please kindly inform us. Thank you very much for your kind consideration.

Best Regards,

Professor Dr. Areerat Suputtitada, MD.
On behalf of the authors

Response to Reviewer 3 Comments
Thank you for revising the manuscript. It was improved. Please find my follow comments: 

•    The following sentences need supportive references

''Hemiplegia is one of the most prevalent disabilities seen in the poststroke phase and is characterized by 47 asymmetrical gait abnormalities. Muscle weakness can cause an asymmetrical gait, which can lead to ineffective 48 movement, a loss of balance, and the risk of musculoskeletal injuries in healthy limbs. Spasticity is a typical 49 symptom of neurological impairment. It is commonly found after a stroke, causing difficulties with everyday 50 activities and a lower quality of life ''.

Response 1 
Hemiplegia is one of the most prevalent disabilities seen in the poststroke phase and is characterized by asymmetrical gait abnormalities. [1,2] Muscle weakness can cause an asymmetrical gait, which can lead to ineffective movement, a loss of balance, and the risk of musculoskeletal injuries in healthy limbs. [2] Spasticity is a typical symptom of neurological impairment. It is commonly found after a stroke, causing difficulties with everyday activities and a lower quality of life.[2]
•    Please link the followinf paragraph with the limitation in the current evidence: 
'' Electromyography (EMG) is used to predict poststroke gait and rehabilitation therapy because it is sensitive to 82 neuromuscular abnormalities produced by stroke. [27-34] For the development of an EMG-based static and 83 dynamic spasticity monitoring system, we proposed a portable EMG device or dynamic electromyography 84 (dEMG), data processing, and data analytics for convenience in clinical practice. The dEMG is an objective 85 spasticity measurement. The Modified Ashworth Scale shows a good relationship with the amplitude and duration 86 of the dEMG response in the stretch reflex manoeuvre as a measure of static spasticity. The walking root mean 87 square (RMS) amplitude and duration are said to be linked to dynamic spasticity when walking. [27-30] Some 88 studies previously investigated the relationship between dEMG and muscle tension and discovered that during 89 voluntary contractions, dEMG and muscle tension exhibited a positive association. [31-33] Another study revealed 90 that dEMG, combined with an isokinetic test, was a good quantitative method for measuring spasticity.[30]''
Response 2 Line 91-95
Electromyography (EMG) is used to predict poststroke gait and rehabilitation therapy because it is sensitive to neuromuscular abnormalities produced by stroke. [27-34] For the development of an EMG-based static and dynamic spasticity monitoring system, we proposed a portable EMG device or dynamic electromyography (dEMG), data processing, and data analytics for convenience in clinical practice. The dEMG is an objective spasticity measurement. The Modified Ashworth Scale shows a good relationship with the amplitude and duration of the dEMG response in the stretch reflex manoeuvre as a measure of static spasticity. The walking root mean square (RMS) amplitude and duration are said to be linked to dynamic spasticity when walking. [27-30] Some studies previously investigated the relationship between dEMG and muscle tension and discovered that during voluntary contractions, dEMG and muscle tension exhibited a positive association. [31-33] Another study revealed that dEMG, combined with an isokinetic test, was a good quantitative method for measuring spasticity.[30] Since the MAS and Tardieu scale are more subjective and less quantitative, dEMG is considered the objective test of spasticity which is highly accurate. [27-33] However, dEMG measurement necessitates the use of professionals and machines, and it is still underutilized in neuromuscular disease research. In the future, more modern dEMG machines that are simple to use and interpret should be available.

Line 392-395
This is, however, a prospective randomized crossover trial involving only one institution. Additional prospective RCT trials with a larger sample size and a longer follow-up time should be performed in an international, multicentre environment. More research is needed to establish the sensitivity of dEMG in measuring spasticity. Increased computing power, wearable sensors, and mobile connectivity have launched in plenty of mobile health innovations that have the potential to improve stroke detection and EMS response. Future studies should combine dEMG with cutting-edge technologies such as Automatic Acute Stroke Symptom Detection and Emergency Medical System Alerting via Mobile Health Technologies. [35] This study is unique and interesting since the AAFO is easy to produce and inexpensive, and dynamic electromyography devices inexpensive machines that should be recommended in clinical practice due to their portability, light weight, and ease of use by the clinician. This study could be used as a foundation for future global research.

- The manuscript is still very long and hard to follow.
Response 3. The manuscript was edited and submitted for the editing service in English language from Springer Nature with a certificate attached.
